# VIOLENCE DETECTION AND LOCALIZATION IN VIDEO THROUGH SUBGROUP ANALYSIS

## ABSTRACT

In an era of rapid technological advancements, computer systems play a crucial role in early Violence Detection (VD) and localization, which is critical for timely human intervention. However, existing VD methods often fall short, lacking applicability to surveillance data, and failing to address the localization and social dimension of violent events. To address these shortcomings, we propose a novel approach to integrate social subgroups into VD. Our method recognizes and tracks subgroups across frames, providing an additional layer of information in VD. This enables the system to not only detect violence at video-level, but also to identify the groups involved. This adaptable add-on module can enhance the applicability of existing models and algorithms. Through extensive experiments on the SCFD and RWF-2000 surveillance datasets, we find that our approach improves social awareness in VD by localizing the people involved in an act of violence. The system offers a small performance boost on the SCFD dataset and maintains performance on RWF-2000, reaching 91.3% and 87.2% accuracy respectively, demonstrating its practical utility while performing close to state-of-the-art methods. Furthermore, our method generalizes well to unseen datasets, marking a promising advance in early VD.

## 1 INTRODUCTION

Surveillance cameras are ubiquitous in modern society, and they play a crucial role in maintaining public safety. However, the sheer volume of footage captured by these cameras can be overwhelming for human analysts to review. CCTV operators face multiple challenges in their work such as a high camera to operator ratio, distractions in the workplace, and long working hours (Hodgetts et al., 2017; Keval & Sasse, 2010), making it difficult to concentrate on all cameras at all times. As a result, there is a pressing need for automated tools that can quickly and accurately detect but also localize instances of violence in surveillance footage. Such a system would benefit from providing meaningful results, explicitly including individuals and the groups they are a part of.

Existing methods for Violence Detection (VD) often depend on training data that is staged or contains very specific situations such as hockey fights or kickboxing (Bermejo Nievas et al., 2011; Kwan-Loo et al., 2022), with larger surveillance datasets emerging in recent years (Aktı et al., 2019; Cheng et al., 2021). Additionally, the focus lies either on detection but not localization, or localization is not explicitly based on persons or groups, leaving a gap between frameworks and interpretable real-world applications.

We therefore propose to incorporate subgroups into the task of VD, serving two purposes: it will increase interpretability of the outcome, by indicating where violence was detected, as well as increasing appropriateness for safety systems by enabling tracking and analysis of subgroups for VD. Furthermore, since it is an add-on module, it can be combined with any model or system. Acknowledging the importance of both interpersonal (Rota et al., 2015) and contextual (Freire-Obregón et al., 2022) information, we combine full video analysis with subgroup analysis, automatically extracting and tracking subgroups.

Our contributions are as follows:

- We propose the integration of a subgroup analysis module into violence detection methods, addressing the need for increased interpretability and applicability of safety systems.

- This subgroup module detects and localizes violent events by automatically extracting and tracking groups across frames, while enabling a slight improvement or maintained performance on the overall task.

- The system proves to generalize well to unseen surveillance data, furthermore underlining its utility for safety systems working with real-world data and its potential to reduce the workload of human analysts.

The remainder of this paper is structured as follows. A brief summary of related work is provided in Section 2, followed by an overview of the methods employed in Section 3. Experiments and their results are discussed in Section 4, after which the paper is concluded in Section 5.

## 2 RELATED WORK

### 2.1 VIDEO VIOLENCE DETECTION

The VD research field is rapidly expanding, with recent developments including the creation of specialized datasets for violence recognition from surveillance footage. In 2019, Aktı et al. (2019) proposed the SCFD dataset, together with a framework using a CNN for feature extraction and a bidirectional LSTM with an attention layer for classification. Cheng et al. (2021) introduced the RWF-2000 dataset and a pipeline combining RGB and optical flow data, emphasizing movement in RGB areas. Optical flow is also utilized in Ullah et al. (2022b) for CNN-based anomaly detection in IoT environments, whereas Islam et al. (2021) work with cost-effective alternatives of optical flow. Another approach for VD on small devices is described in Vijeikis et al. (2022), combining a spatial feature extractor with an LSTM temporal feature extractor. Kang et al. (2021) apply 2D CNNs, merging three consecutive frames into one by averaging the RGB channels per frame, with a temporal attention module. The work of Tan & Liu (2022) proposes to employ anomaly detection to find training data for supervised action recognition, thereby using both tasks iteratively. Su et al. (2022) highlight the importance of hyperparameter tuning, with an efficient general action recognition CNN outperforming techniques created specifically for VD. Kwan-Loo et al. (2022) track individuals across frames by finding the largest Intersection over Union of bounding boxes, combining pose information from past and current frames for classification.

### 2.2 VIOLENCE LOCALIZATION

While most studies primarily address VD, recent research has shown a growing interest in violence localization. In Roman & Chávez (2020), a masking model generates motion saliency masks from dynamic images, merging salient regions near detected individuals to identify the main violent area. Mohammadi & Nazerfard (2023) employ a reinforcement learning model to assess the significance of RGB and optical flow frame regions, recursively cropping the highest-scoring area for final classification. This is done recursively, classifying the final cropped region. Asad et al. (2022) use spatiotemporal attention modules and bidirectional convolutional LSTMs to learn masks for each video, creating a heatmap overlay indicating important classification regions. Something similar is done in the previously discussed Su et al. (2022) and Kang et al. (2021), where Grad-CAM (Selvaraju et al., 2017) is employed to visualize the regions on which the classification model focuses.

### 2.3 SUBGROUP ANALYSIS FOR VD

Some research has explored subgroups within VD. Chang et al. (2010) track individuals using multiple cameras for non-violent group actions, such as the formation of groups, grouping them per frame. VD is a separate component, for which motion features are extracted from the foreground of frames and fed to an SVM for classification. Freire-Obregón et al. (2022) investigate the influence of context by tracking individuals, and applying a threshold to determine the percentage of overlap the bounding boxes of two people should have to not be removed as background, effectively grouping people together. Finally, in the work of Rota et al. (2015), violence is both detected and localized by only considering movements happening in the space between two people. When criteria for this interpersonal movement are met, visual features are classified with an SVM.

## 2.4 COMPARING PREVIOUS WORK TO THE PROPOSED FRAMEWORK

Most studies on surveillance VD do not incorporate localization, making the output less interpretable (Aktı et al., 2019; Cheng et al., 2021; Islam et al., 2021; Kang et al., 2021; Ullah et al., 2022b; Su et al., 2022; Tan & Liu, 2022; Vijeikis et al., 2022). Those who do either provide heatmaps that are not guaranteed to have social meaning (Asad et al., 2022; Kang et al., 2021; Su et al., 2022), or they are unable to point out more than one violent area (Asad et al., 2022; Mohammadi & Nazerfard, 2023). While such heatmaps serve interpretability, they are not guaranteed to include people or to localize multiple distinct regions. Research involving explicit group inclusion either groups individuals per frame using multi-camera tracking (Chang et al., 2010) or focuses solely on groups or the entire video, without combining these aspects (Freire-Obregón et al., 2022; Rota et al., 2015). Bridging these gaps, our proposed system detects and localizes violence from single-camera surveillance footage, by incorporating the full video as well as cropped subgroups from that video. This enables generation of a socially-aware output, where multiple subgroups are tracked throughout the video and classified as depicting violence or not.

## 3 METHODOLOGY

An overview of the model is presented in Figure 1. Each video serves as input for two streams: one for full-video violence recognition, and one for subgroup violence recognition. For the latter, location and optical flow information are extracted as features for each detected person. These features, and thus the individuals they are retrieved from, are then clustered per frame. People are tracked across frames, to go from frame-level subgroups to video-level subgroups. Each subgroup is then fed to a VD network, the output being fused with the violence prediction of the entire video.

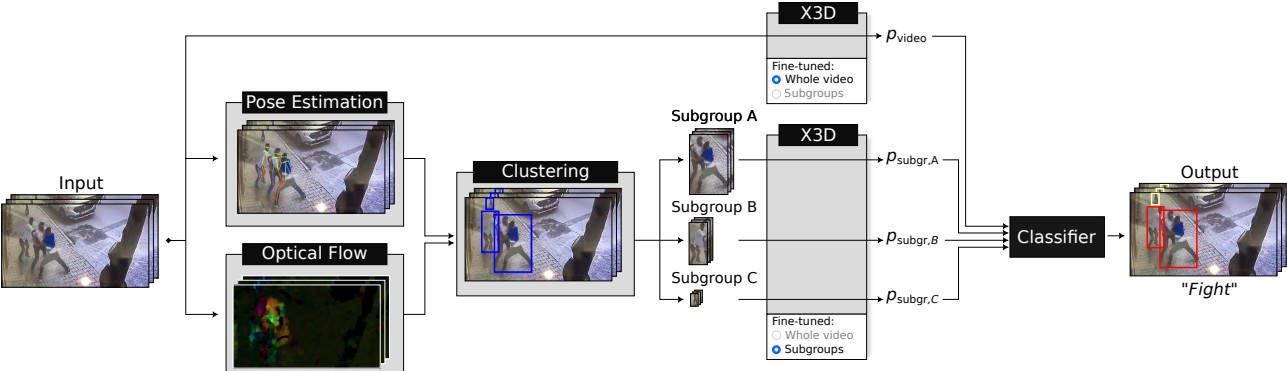

Figure 1: Overview of the proposed method.

## 3.1 GENERAL DESIGN CHOICES

Incorporating information from both the entire scene and cropped subgroups within a video is motivated by their individual significance in VD (Freire-Obregón et al., 2022; Rota et al., 2015), and the interpretability gained through subgroup-based analysis. Analyzing groups, instead of individuals, is chosen due to previous work showing that visual information from the interpersonal space of multiple people greatly contributes to behavior classification (Rota et al., 2015). Furthermore, it is worth noting that analyzing solitary individuals adds little value to the analysis of individuals close together (Freire-Obregón et al., 2022), since violence tends to involve multiple people. We focus on detection over prediction to avoid prediction bias, a serious issue where present inequalities are projected onto the future (Kang et al., 2021; Mayson, 2019). Our subgroup detection approach is similar to that of Veltmeijer et al. (2022), with the main difference being our use of motion features instead of face features, reducing privacy concerns and potential data biases (Kang et al., 2021).

## 3.2 DATA

The model was trained and evaluated on two datasets, the Surveillance Camera Fight Dataset (SCFD) (Aktı et al., 2019) and the Real-World Fighting (RWF-2000) dataset (Cheng et al., 2021), two of the main violence datasets that consist of general surveillance footage only (Ullah et al., 2023). SCFD contains 300 surveillance video clips, 150 of which contain violence and 150 not. Each clip is 2 seconds long. RWF-2000 contains 2000 surveillance video clips, 1000 violent clips and 1000 non-violent clips, each clip lasting 5 seconds. Since both datasets mainly contain physical fights, the words 'violence' and 'fight' will be used interchangeably in the remainder of this work.

We extend the existing dataset annotations, by assigning labels to subgroups from videos in the fight classes. Since the non-fight videos are already labeled as such, we assume that none of the videos nor subgroups will contain fights. For the fight videos, on the other hand, individual subgroups are not guaranteed to contain fights. A label of fight or non-fight is assigned to each individual subgroup detected within that video. The SCFD subgroups are annotated by two different annotators. From this an inter-rater reliability (IRR) is calculated, giving a Cohen's kappa value of $\kappa$=0.90(Cohen, 1960). This indicates an 'almost perfect'(Landis & Koch, 1977) or 'strong' (McHugh, 2012) agreement, allowing for the remainder of data being assigned to one of the annotators. This results in 94 (SCFD) and 481 (RWF-2000) subgroups being labeled as violent, versus 205 (SCFD) and 594 (RWF-2000) labeled as non-violent.

Since the network requires a square input, data is resized in two different ways: center cropping and zero padding, both combined with resizing the resulting square frame to 160×160. For the full videos, all training data is augmented in both ways, resulting in two square videos per original video. SCFD results are reported on test data that is cropped, RWF-2000 results that is padded, generalization results on both combined. Subgroup videos from both datasets are padded only, so as not to further crop the already cropped subgroup. The only other data augmentation applied is flipping, which is done for the subgroups videos (SCFD) and full videos and subgroup videos (RWF-2000) of the training sets.

## 3.3 FRAME-LEVEL SUBGROUP FORMATION

### 3.3.1 FEATURE EXTRACTION

For each video, we extract frames and dense optical flow information (pixel-wise angle and magnitude) using OpenCV (Bradski, 2000) and sample eight frames per video. Pose estimation is performed on the remaining frames using AlphaPose (Fang et al., 2022), a recognized framework for precise and robust keypoint detection (Inturi et al., 2023; Zwölfer et al., 2023). Frames are retained if they contain detected individuals with assigned poses.

We utilize the pose estimation's inferred center coordinates of bounding boxes as location features, following the methodology of Veltmeijer et al. (2022). We aim to extract meaningful motion features, where individuals moving in a similar direction have more similar motion features. To achieve this, we calculate the average gradient over all pixels within each individual's bounding box in the optical flow output as the motion feature.

### 3.3.2 CLUSTERING

Following the procedure of Veltmeijer et al. (2022), we employ hierarchical clustering for forming meaningful subgroups from the individuals in each frame. The coordinates of each person together with the movement information form the elements of the feature vector to be used for clustering. The feature vector of each individual consists of three elements: $[x, y, gradient]$. $x$ and $y$ are normalized by dividing them by the image size in their respective dimension, and the gradient value is converted to range from 0 to 360 degrees. The optimal number of clusters for each frame is determined from the resulting dendrogram in an automated fashion.

For each cluster, we save a bounding box encompassing all individuals within that cluster, based on their original pose estimation bounding boxes. To ensure accuracy for training, a bounding box should only contain individuals assigned to that specific subgroup. Therefore, if one subgroup is interrupted by another, we split the interrupted subgroup to eliminate the interruption, focusing on

interruptions along the horizontal axis (x-values). Specifically, cluster $A$ is split up when a cluster $B$ exists with:

$$B_x \subset A_x \tag{1}$$

In this case cluster $A_x$ is split up into clusters recursively until none of the from $A_x$ derived clusters are a superset of any other cluster.

### 3.4 VIDEO-LEVEL SUBGROUP FORMATION

So far, we discussed person detection and subgroup formation at frame-level. The next step is to track individuals throughout the video and find video-level subgroups. Establishing video-level subgroups based on the clustered subgroup per frame will smooth frame-level predictions. While Alphapose (Fang et al., 2022) offers tracking options, we find them to be unstable for our purpose due to temporal consistency issues. Tracking methods often involve calculating the smallest distance in a person's location between two frames (Chang et al., 2010; Kwan-Loo et al., 2022). However, in scenes with clutter or fast movements, often encountered in fight scenes, this can result in frequent mismatches. Consequently, tracking and analyzing movement in violent scenarios remain challenging, with existing algorithms frequently falling short (Rota et al., 2015; Ullah et al., 2023).

Instead, in this work individuals are tracked by predicting their future location and finding the closest match, if possible, to that predicted location. The center coordinates of an individual's bounding box are used as location, rather than the (size of the) bounding box itself, to increase stability. For all individuals with center location $C_n = (x_n, y_n)$ in frame $n$, optical flow gradient and magnitude are used to calculate the predicted center location $P_{n+1} = (x_{n+1}, y_{n+1})$ of that individual in frame $n+1$. A distance matrix $M_{\text{dist}}$ contains the Euclidean distance between the predicted center location $P_{n+1} = (x_{n+1}, y_{n+1})$ of individuals detected in frame $n$ and the actual center location $C_{n+1} = (x_{n+1}, y_{n+1})$ of individuals detected in frame $n+1$. The distance matrix is then used for matching individuals between frame $n$ and frame $n+1$, by solving it as a linear sum assignment problem, implemented with SciPy (Jonker & Volgenant, 1988; Virtanen et al., 2020). It is possible that an individual in frame $n$ is not detected or present in frame $n+1$, or vice versa. To account for this, the distance matrix is padded with a threshold value, to ensure that individuals that have no plausible match will be matched to a padded value. This threshold value is set to 40, which is empirically found to generally distinguish between the same person moving and two people close together moving. When no match is found for an individual in frame $n+1$, a match is sought with a non-matched individual in frame $n-1$. In this case, the threshold is set to 50, meaning that a match can only be made if the Euclidean distance is below 50. If an individual is not matched for two consecutive frames, the tracking of that individual stops. In the end, only tracked individuals that have been identified in at least four of the eight frames are further considered.

After tracking and grouping individuals per frame, we use pairwise majority voting to infer video-level subgroups, in a way similar to Veltmeijer et al. (2022). Instead of finding consensus across multiple subgroup annotations for the same image, we propose to reach subgroup consensus over multiple frames. Following their approach, we consider a pair of individuals to be in the same video-level subgroup if they are part of the same frame-level subgroups for a majority of frames This is visualized in Figure 2.

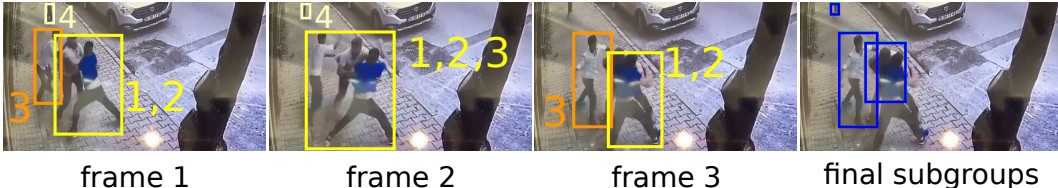

| frame 1 | frame 2 | frame 3 | final subgroups |

Figure 2: Approach for merging subgroups across frames. Person 1 and 2 are in the same subgroup for a majority of frames, and are therefore put together in the same video-level subgroup. Person 3 and 4 are on their own for a majority of frames, the latter not being detected in frame 3.

## 3.5 Network training

For the detection of violence in both the full videos and the subgroup videos, we employ the X3D network (Feichtenhofer, 2020), an efficient network for video recognition, as adjusted and provided by Su et al. (2022); Su (2022) and pretrained on the kinetics dataset (Kay et al., 2017). For training, we leave most hyperparameters as they are, changing only the values of *clip_len* (8), *alpha* (1) and *tau* (1). The number of training epochs is changed from 1 to 10, saving the model after the epoch in which it performs best on the test set. The learning rate is kept at $1 \times 10^{-4}$. The network is trained separately for detecting violence in the full videos and for detecting violence in subgroup videos.

For SCFD, we perform 5-fold cross-validation, using the same fold division as Su et al. (2022) to ensure a fair comparison. For this division, videos were randomly divided into the folds while ensuring a fair balance between fight and non-fight clips, and assigning all videos originating from the same original (longer) video to the same fold to avoid data leakage. For RWF-2000, we use the training and test sets as originally provided by Cheng et al. (2021), which also prevent data leakage by assigning all clips from the same original video to the same fold.

## 3.6 Prediction fusion

The prediction resulting from the model trained on the full videos serves as a baseline, and as the foundation on which the trained subgroup model should build. Since not all people are recognized or tracked successfully, the absence of violent subgroups is no reliable indicator of the original video to not depict any violence. The opposite, however, holds: once one of the subgroups from a video is classified as showing violence, we hypothesize that this is a strong indicator for the full video to depict violence (Mohammadi & Nazerfard, 2023). A non-fight prediction for a video, $y_{\text{video}} = 0$, can therefore change to a fight prediction, $y_{\text{video}} = 1$, if the probability $p_{\text{subgr}}$ of at least one of the subgroups from that video showing violence is 0.8 or higher. This means that

$$y_{\text{video}} = \begin{cases} 1, & \text{if } p_{\text{video}} \geq 0.5 \text{ or } \text{maxProb}(p_{\text{subgr}}) \geq 0.8 \\ 0, & \text{otherwise.} \end{cases} \tag{2}$$

## 4 Experiments

### 4.1 SCFD

This section will describe and discuss the performed experiments and their results on SCFD, the main dataset under analysis on which most experiments are performed. For the first baseline, violence detection is performed based on the full video analysis only, results of which are given in the first row of Table 1. These values show that the network is already performing quite well on the task of VD in the SCFD dataset, which is similar to the results and findings presented in Su et al. (2022).

For the next experiment, the network is not trained on full videos, but only on subgroup videos for violence detection. Results are presented in the second row of Table 1. Note that these performances cannot be directly compared to those of the full model, since a subgroup originating from a 'fight' video might be part of the 'non-fight' subgroup set when none of the individuals in that subgroup are displaying violent behaviour.

To validate that any performance gain obtained from combining the full video with subgroup prediction would be a result of the subgroups themselves, rather than simply resulting from analysing separate crops of the video, we perform a final baseline experiment. For this experiment we replace each subgroup video with a random crop video of the original video, with the same size as the original subgroup video. The best performing trained subgroup model is then employed to evaluate these random rectangle videos, and the results are combined with the full video predictions as described in 3.6. The fold-wise results of this are included in Table 1, showing that the outcome is the same as the baseline model, which suggests that the improved performance of the subgroup model is not a result of cropping parts of the video in itself, but of what is present in those crops.

Finally, full video predictions are combined with subgroup predictions in the way described in Section 3.6, completing Table 1. These results show that adding subgroups to the full video analysis

Table 1: Weighted accuracy per fold and average weighted accuracy and F1-score of the baseline model, random rectangle model, subgroups only model, and full video and subgroups combined.

|  | fold 1 | fold 2 | fold 3 | fold 4 | fold 5 | avg | F1 |
|---|---|---|---|---|---|---|---|
| full video | 91.3% | 92.1% | 91.7% | 83.3% | 91.5% | 90.0% | 0.90 |
| subgroups | 85.3% | 85.5% | 82.6% | 84.4% | 81.1% | 84.1% | 0.85 |
| full video + rectangles | 91.3% | 92.1% | 91.7% | 83.3% | 91.5% | 90.0% | 0.90 |
| full video + subgroups | 93.1% | 93.7% | 91.7% | 86.7% | 91.5% | 91.3% | 0.91 |

improves performance for three of the folds, and performs on par with the full video only for two of the folds. When considering the violent class only, performance increases more, but this is tempered by an increase of false positives decreasing the performance of non-violence. Furthermore, adding random rectangles to the full video analysis instead has no influence on the final result whatsoever. This makes clear that even though the overall increase in accuracy for the proposed model is quite small, it is consistent throughout different folds and can be attributed to the nature of the subgroups, therefore being a meaningful addition. Output of the system is visualized in Figure 3, depicting violent subgroups with red bounding boxes.

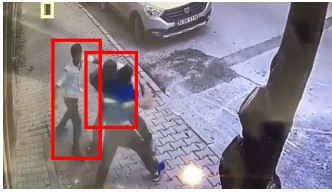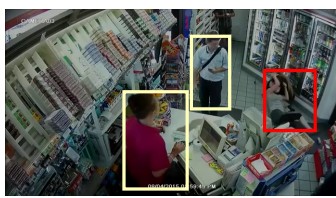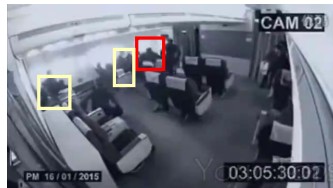

Figure 3: Output of the proposed method, violent subgroups are indicated with red bounding boxes.

## 4.2 RWF-2000

This section will describe and discuss the experiments on RWF-2000. This dataset is used for validating the model and testing the generalizability. To validate the proposed system, it is also trained and tested on the RWF-2000 surveillance dataset. Results are presented in Table 2, indicating a minuscule difference in performance between the baseline and the proposed model, the addition of subgroups leading to a performance loss of 0.3%. Since the original dataset comes with a fixed training and testing splits, no cross-validation is performed to enable valid comparisons to other frameworks. It should be noted that the proposed architecture was built for SCFD, as the main dataset used in this work. This could explain that the performance gain found for SCFD does not extrapolate when training and testing on RWF-2000. Furthermore, it should be noted that the RWF-2000 dataset is generally considered to be more challenging. Not only because of the highly variable resolution of the surveillance footage, which can also be said for SCFD, but also due to some videos being modified versions rather than raw surveillance footage (Vijeikis et al., 2022). Furthermore, while annotating the subgroups, the researchers noticed that the video-level annotations are not fully correct: all clips originating from the same video have been given the same video-level annotation of violence, while clips from the start or end of the original video do not show violence themselves. No video-level annotations were changed for the sake of comparison, but future work could analyse this to further improve this important and relevant dataset.

## 4.3 BENCHMARK RESULTS

Performance of our proposed combined model is compared to the baseline and to other benchmark results in Table 3. It can be seen the baseline model, similar to the implementation of Su et al. (2022), already scores relatively high. For both datasets, multiple clips in the dataset originate from the same original video, meaning that a random split is likely to contain clips from the same video in both the training and test set. This is called data leakage, and causes the model to overfit, thereby

Table 2: Average weighted accuracy and F1-score of the full video only, subgroups only, and full video + subgroups prediction on the RWF-2000 dataset.

|  | $\mathbf{acc}_w$ | **F1** |
|---|---|---|
| full video only | 87.5% | 0.87 |
| subgroups only | 75.4% | 0.75 |
| full video + subgroups | 87.2% | 0.86 |

Table 3: Accuracy of previously published models on violence detection, as reported in their respective papers, compared to our baseline and proposed subgroup model. Models under the line, marked with an *, were trained and tested on the same data splits, and that there was no data leakage.

| method | SCFD | | RWF-2000 | |
|---|---|---|---|---|
|  | acc | F1 | acc | F1 |
| Aktı et al. (2019) | 72.0% | - | - | - |
| Ullah et al. (2022a) | 74.0% | 0.73 | - | - |
| Vijeikis et al. (2022) | - | - | 82.0% | 0.78 |
| Ullah et al. (2021) | 75.9% | 0.75 | 88.2% | 0.89 |
| Kang et al. (2021) | 92.0% | - | - | - |
| Tan & Liu (2022) | 95.6% | - | - | - |
| Islam et al. (2021) | - | - | 89.8% | - |
| Ullah et al. (2022b) | - | - | 93.3% | - |
| Transfer model baseline (ours) | 86.3% | 0.86 | 83.3% | 0.83 |
| Transfer model proposed model (ours) | 88.3% | 0.88 | 82.4% | 0.82 |
| Cheng et al. (2021) | - | - | 87.3%* | - |
| Mohammadi & Nazerfard (2023) | - | - | 90.4%* | - |
| Kang et al. (2021) | - | - | 92.0%* | - |
| Su et al. (2022) | 88.7%* | - | **94.0%*** | - |
| Baseline (ours) | 90.0%* | 0.90* | 87.5%* | **0.87*** |
| Proposed model (ours) | **91.3%*** | **0.91*** | 87.2%* | 0.86* |

giving a distorted view of the actual performance of the model. It should be noted that for SCFD, Su et al. (2022) are the first and (to the best of our knowledge) only ones to have explicitly mentioned this, splitting the dataset into five balanced splits actively preventing data leakage. We have used their division into folds. This means that for none of the other works in Table 3 reporting on SCFD, we can state with certainty that there was no data leakage, therefore our work can only fairly be compared to that of Su et al. (2022). This is indicated in Table 3 by placing comparable works under the horizontal line. We encourage researchers to use the same fold division and share more information on the data splits used, for fair comparison as well as data leakage prevention. RWF-2000 was published with a predefined training and test set. Those papers that mention using this split, as was done in this study, are placed under the horizontal line in Table 3.

## 4.4 GENERALIZABILITY TO UNSEEN DATASETS

For a VD framework to be applicable in real-life scenarios, it is important to be robust to highly variable inputs. Specifically, this means that it should be able to generalize to unseen surveillance data. To test the generalizability of the proposed framework, we perform two experiments: training on SCFD data and testing on RWF-2000 data, and vice versa. Results are presented in the two rows just above the line in Table 3, showing that the model generalizes exceptionally well. Trained on (all of) RWF-2000 and tested on (all of) SCFD, we see that accuracy even increases with 2% when adding the subgroup module, to a decent score of 88.3%. The model that is trained on (all of) SCFD and tested on (all of) RWF-2000 shows a pattern similar to that of the model both trained and tested on RWF-2000, in that addition of the subgroup model leads to a slight drop in performance. However, it is noteworthy that even when trained on the relatively small SCFD, performance on

RWF-2000 still reaches an accuracy of around 83%. It should be noted that our transfer models do not fully meet the aforementioned criteria of data leakage prevention: they are trained on one entire dataset and tested on the other entire dataset. They are therefore placed above the horizontal line.

## 5 CONCLUSION

In this paper, an innovative approach for incorporating subgroup analysis into VD was proposed. While there is a vast body of research on VD, little work has been done on socially meaningful and interpretable VD in surveillance footage. Our adaptable add-on module automatically extracts and tracks subgroups across frames for violence detection and localization in safety systems. As such, it can help bridge the gap between theory and practice, by alerting when violence is detected and indicating what group of people is or are involved.

We trained an efficient network, X3D, and performed experiments on two of the main violence surveillance datasets: SCFD and RWF-2000. Results indicate that our subgroup module consistently increases or stabilizes the performance on SCFD with on average +1.3%, while leading to a small decrease in performance on RWF-2000 (-0.3%), needing as little as eight frames per video. In the proposed system, subgroup analysis can only change a non-violent label to a violent one, increasing true positives and false positives for the violence class. In practical use cases, we recommend prioritizing false positives over false negatives since having to rewatch a fragment will have fewer consequences than missing a violent event. Models trained on either dataset demonstrate strong generalizability to the other, promising broader applicability. Figure 3 illustrates model outputs, showing meaningful and actionable information for CCTV operators. These results suggest the framework's potential to enhance public safety and reduce human analyst workload.

A main limitation is that subgroup formation and therefore system performance rely heavily on the initial pose estimation. Individuals without assigned keypoints will not be part of any subgroup and thus not influence the final prediction. This poses challenges in VD, as fighting individuals are often blurred, occluded, or in less detectable positions. We see that missed keypoints are typically associated with individuals engaged in fights, which are crucial for violence detection. This led to the decision of only allowing the module to change a non-fight label to a fight-label, as the absence of detected fighting subgroups may indicate missed individuals rather than their absence in the entire video. Improving person detection could pave the way for a more refined subgroup analysis influencing the overall prediction.

While the current work focused on building the subgroup module and how it influenced the baseline, reaching state of the art performance was not a main motive. Future work should experiment with adding this module to other frameworks, including those outperforming the models described here. Another direction for future work is to include temporal attention for detecting which frames of a video are most important for the final classification. While our system could readily be extended to this, by feeding it multiple blocks of eight consecutive frames from the same video, integrating it into the framework would further enhance its practical utility.

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
