# OpenReview forum: "Violence Detection and Localization in Video Through Subgroup Analysis"
_ICLR.cc/2024/Conference — ICLR 2024 Conference Withdrawn Submission_

### Official Review · Reviewer_WPpq · 2023-10-26

**Soundness:** 2 fair
**Presentation:** 2 fair
**Contribution:** 3 good
**Rating:** 3
**Confidence:** 4

**Summary:**

Violence Detection (VD) deals with the early detection and localization of violent events to enable timely human intervention. However, existing VD methods have their limitations, especially in processing surveillance data and considering the localization and social aspects of violent events. To overcome these shortcomings, the authors propose an innovative approach that incorporates social subgroups into VD.

Their method involves detecting and tracking subgroups across frames, adding an additional layer of information to VD. This allows the system to not only detect violence at the video level, but also identify the groups involved. This adaptable add-on module extends the applicability of existing VD models and algorithms.

The authors conducted extensive experiments on the SCFD and RWF-2000 surveillance datasets. The results show that their approach improves social awareness in video surveillance by accurately locating individuals involved in violent crimes. The system achieved a small performance improvement in the SCFD dataset and maintained its performance in the RWF-2000 dataset, achieving 91.3% and 87.2% accuracy, respectively. Importantly, the approach also showed good generalization to unseen datasets, representing a promising advance in the early detection of violence.

**Strengths:**

.) The paper is well structured and well written, the contribution is clear
.) The experimental section and comparision with SOTA is very good.
.) The introduction of subgroups provides an improvement of the overall analysis task.

**Weaknesses:**

One of the main goals is that the results enhance public safety, but this is not shown at all. The paper provides a good technical contribution but fails to address the social dimensions as all the others briefly mentioned in the abstract.
Section 3.2: The paper contains assumptions what violence is, but does not provide a definition. It s also unclear how violence is defined in the datasets.
No tests in real scenarios, but it is claimed that the system works in real world.
The main novelty is the introduction of sub groups who have a marginal impact on the overall outcome. So innovation is rated rather limited.

**Questions:**

3.2: what is the content of the datasets? What do they show? How is Violence defined: it seems violence is defined as what is annotated in the video data? Did you investigate any biases in the dataset?
Provide a precise definition of violance!

**Details Of Ethics Concerns:**

Detecting violence without any definition what it is, is not a scientific ethical approach. Claiming that this enhances public safety without any prove is also not a scientific/ethical approach. Analysis of people s behaviour and classifying between violent and non violent people might discriminate people, might be against GDPR, and needs an ethical statement. Moreover the underlying dataset is not investigated with respect to existing bias, or fairness constraints.

---

### Official Review · Reviewer_n6jP · 2023-10-31

**Soundness:** 2 fair
**Presentation:** 3 good
**Contribution:** 2 fair
**Rating:** 3
**Confidence:** 4

**Summary:**

This paper proposes a system for violence detection and localization from surveillance videos. The proposed method extracts pose info and optical flow from the videos and clustering is performed to get potential crops. The crops (action proposals) are fed into an X3D network to extract features for the classifier. Experiments on two surveillance video datasets on detecting the “fight” action class show the efficacy of the proposed method.

**Strengths:**

1. The proposed method achieves SOTA on one dataset.
2. The paper is generally well-written and easy to follow.

**Weaknesses:**

1. There is no inference speed comparison. The model uses pretrained pose estimation models and needs to extract optical flow, which is computationally expensive. In real-world applications, surveillance video action detection systems will be required to run in real-time on certain hardware configurations. Therefore a speed analysis (for each component, if possible) is necessary.
2. In terms of the system design, the X3D feature extraction process seems redundant. From Figure 1, the proposed method extracts full video features and subgroup features, which wastes computation on feature extraction on overlapping pixel areas. A more efficient way to extract features is to employ ROIAlign from a single feature map as in [1*, 2*]. Also these key implementation details are missing in the text.
3. This paper is more of a computer vision “system” paper, which is more suitable for a CV conference like CVPR or WACV.
4. Missing references on a series of surveillance video action detection works:
[1*] Argus: Efficient activity detection system for extended video analysis. WACVW 2020.
[2*] Gabriellav2: Towards better generalization in surveillance videos for action detection. WACVW2022

**Questions:**

1. In terms of the evaluation, the authors report accuracy and F1. However, since the task is to output bounding boxes over the violence time interval, some kind of matching between predicted bounding boxes and ground truth is needed (like the IOU threshold and mAP metric in object detection literature). Can the authors elaborate on how this matching is done and why the current metric does not reflect this?

---

### Official Review · Reviewer_u3A5 · 2023-10-31

**Soundness:** 2 fair
**Presentation:** 2 fair
**Contribution:** 2 fair
**Rating:** 5
**Confidence:** 3

**Summary:**

This paper proposes a method to solve the violence detection method. The proposed method proposes a subgroup clustering strategy along with the whole video to detection violence in videos. The proposed method achieves state-of-the-art performance on some benchmark datasets.

**Strengths:**

In general, I think the proposed paper has clear illustrations and explanations. The proposed method is somehow simple and can be easily implemented.

**Weaknesses:**

However, I think this paper has some severe drawbacks:

1. The proposed method somehow lacks novelty here. The proposed method (subgroup and part analysis) in video understanding is not a brand-new idea. Considering [1] and [2], which explicitly split scenes, objects, and persons. I don't think there is a critical difference between those works and the proposed method. Also, I don't think there exists novelty in the clustering part. To me, this paper is more like a technical report rather than a research paper. This is my critical concern.

2. To me, the experiment part needs to be revised. The author does not provide extra ablations to verify the effectiveness of the proposed model like the prediction fusion threshold, subgroup threshold, etc.
1: Choi J, Gao C, Messou J C E, et al. Why can't I dance in the mall? learning to mitigate scene bias in action recognition[J]. Advances in Neural Information Processing Systems, 2019, 32.

2: Wang Y, Hoai M. Pulling actions out of context: Explicit separation for effective combination[C]//Proceedings of the IEEE Conference on Computer Vision and Pattern Recognition. 2018: 7044-7053.

**Questions:**

Please mainly see the weaknesses section for details. Besides those weaknesses section, I have some extra questions:

1. In Section 3.3.1, the author mentioned that they use X3D backbone to extract features. Why does the author choose X3D as the backbone? It is not a popular backbone, especially with recent Transformer models like VideoMAE, MeMViT, etc.

2. In Section 3.3.2, the author mentioned using the linear sum assignment problem with scipy and the center coordination. I think there can be more choices rather than center coordinates, i.e., bounding boxes (with SimOTA or related techniques). I was wondering if a better design will bring better results.

---

### Official Review · Reviewer_bXAa · 2023-10-31

**Soundness:** 3 good
**Presentation:** 3 good
**Contribution:** 2 fair
**Rating:** 5
**Confidence:** 4

**Summary:**

The paper deals with the problem of detecting and localizing violence in videos, through detection and use of shown subgroups in video frames. It can be applied in safety systems in real-world settings, thus reducing workload of human analysts. Experiments are presented on two datasets, SCFD and RWF-2000.

**Strengths:**

The paper extends the ability to detect violence in videos by also considering the existence of cropped subgroups of persons, e.g., fighting, in video frames. Experimental results are shown using two existing image datasets.

**Weaknesses:**

The presented work marginally extends existing frameworks (e.g., Veltmeijer) from a technical point of view.
Moreover, the paper, as also mentioned in the conclusions, does not target achieving a state of the art performance, but states that this is a future work. However, this affects the performance and, thus, the comparisons shown in the Tables (especially 3). As a result, it is not clear what the Tables show, as F1 score and ACC are lower when the proposed subgroup detection is included (in the RWF case).

**Questions:**

Why not including temporal attention in the current paper?

---

### Author Response · Authors · 2023-11-20
**Withdrawal Statement**

We would like to thank the reviewers for their time and constructive feedback. The points raised are truly valuable and will help us further improve our manuscript. We have decided to withdraw the paper, to properly integrate the feedback into our work for a future submission.